# Characterization of the cholangiocarcinoma drug pemigatinib against FGFR gatekeeper mutants

Qianmeng Lin[1,2], Xiaojuan Chen[1,2], Lingzhi Qu[1], Ming Guo[1], Hudie Wei [1], Shuyan Dai[1], Longying Jiang [1✉] & Yongheng Chen [1✉]

Fibroblast growth factor receptor (FGFR) dysregulation is involved in a variety of tumor-igenesis and development. Cholangiocarcinoma is closely related with FGFR aberrations, and pemigatinib is the first drug approved to target FGFR for the treatment of cholangiocarci-noma. Herein, we undertake biochemical and structural analysis on pemigatinib against FGFRs as well as gatekeeper mutations. The results show that pemigatinib is a potent and selective FGFR1–3 inhibitor. The extensive network of hydrogen bonds and van der Waals contacts found in the FGFR1-pemigatinib binding mode accounts for the high potency. Pemigatinib also has excellent potency against the Val-to-Ile gatekeeper mutation but less potency against the Val-to-Met/Phe gatekeeper mutation in FGFR. Taken together, the inhibitory and structural profiles exemplified by pemigatinib may help to thwart Val-to-Ile gatekeeper mutation-based resistance at earlier administration and to advance the further design and improvement for inhibitors toward FGFRs with gatekeeper mutations.

[1] Department of Oncology, Department of Pathology, NHC Key Laboratory of Cancer Proteomics & State Local Joint Engineering Laboratory for Anticancer Drugs, National Clinical Research Center for Geriatric Disorders, Xiangya Hospital, Central South University, Changsha, Hunan 410008, China. [2]These authors contributed equally: Qianmeng Lin, Xiaojuan Chen. ✉email: longyingj1024@163.com; yonghenc@163.com

Fibroblast growth factor receptors (FGFRs) belong to the tyrosine kinase receptor family[1]. It can regulate cell migration, proliferation, and cell differentiation by activating downstream signaling pathways, including the RAS–MAPK–ERK, PI3K–AKT and Janus kinase–signal transducer and activator of transcription (JAK–STAT) signaling pathways[1–4]. Genetic alterations in FGFRs would lead to the aberrations of the activated FGFR signaling pathway and further the establishment and development of a wide variety of cancers[5–8]. FGFR2 aberrations are found in 10–15% of patients with intrahepatic cholangiocarcinoma[9]. Approximately 50% of bladder cancers are associated with FGFR3 mutations, and 30% of hepatocellular carcinoma patients have abnormally increased FGFR4 expression[10,11]. Therefore, targeting the FGFR signaling pathway represents an attractive therapeutic target. Many small-molecule inhibitors are being discovered and approved by the FDA for patients harboring FGFR alterations, such as pemigatinib (Fig. 1A), erdafitinib (Fig. 1B) and infigratinib (Fig. 1C).

Pemigatinib was the first targeted therapy approved by the FDA in April 2020 to treat patients with previously treated, unresectable, locally advanced, or metastatic cholangiocarcinoma and an FGFR2 fusion or other rearrangement[12,13]. In the Phase II FIGHT-202 trial (NCT02924376), pemigatinib yielded an objective response rate of 35.5% (2.8% complete responses) and a disease control rate of 82.0%[14]. These encouraging outcomes underlie the potential benefit of pemigatinib in clinical use. It is supposed that pemigatinib would become a first-line treatment for these patients. An ongoing clinical trial, FIGHT-302, will evaluate the efficacy and safety of pemigatinib therapy versus gemcitabine plus cisplatin combination therapy[15]. Additionally, pemigatinib is expected to treat other FGFR-driven malignancies, such as urothelial carcinoma, and relevant clinical trials are launched and undergoing in various countries worldwide[12].

After an initial response to tyrosine kinase inhibitor therapies, acquired drug resistance gradually develops, correlated with therapy discontinuation and cancer advances, which can be mediated by the alternation in the protein-drug binding and/or the activation of bypass signaling[16,17]. Gatekeeper mutations in kinase domains represent a common theme, which maps at the beginning of the hinge region, curbs the accessibility of the inhibitor to the ATP pocket and contributes to the loss of effectiveness of the inhibitor, such as Abl T315I and EGFR T790M[18–21]. Ponatinib and osimertinib, third-generation tyrosine kinase inhibitors, were developed to overcome Abl T315I and EGFR T790M, respectively[22,23]. Therefore, overcoming resistance by gatekeeper mutations is a novel and subsequent direction of drug design and improvement for next-generation FGFR inhibitors. As the first-generation FGFR inhibitor, pemigatinib needs further development so deeper insights into the inhibitory and structural profiles of pemigatinib are necessary.

In this study, we aimed to explore the structural basis for the high potency of pemigatinib against FGFRs and predict the sensitivity to gatekeeper mutations. We performed biochemical and structural analysis on pemigatinib against FGFRs and gatekeeper mutations. The results show that pemigatinib is capable of inhibiting FGFR1-3 and retains excellent potency against FGFR2 V564I but lower potency against FGFR4 and other gatekeeper mutations. Our results may provide some structural basis and design directions for further optimization of FGFR inhibitors.

## Results

**Potent inhibition of FGFRs by pemigatinib**. To confirm the inhibitory effect of pemigatinib on FGFR1-4, we performed kinase activity inhibition assays. As shown in Fig. 1, pemigatinib exhibits potent inhibitions against FGFR1-3 with half maximal inhibitory concentrations ($IC_{50}$) of 0.2, 1.2, and 1.4 nM, respectively. Compared with FGFR1-3, the inhibition of FGFR4 is greatly reduced (~75-fold less potent compared with FGFR1), with an $IC_{50}$ of 15 nM. We also carried out cellular proliferation assays in Ba/F3 cells to further verify the inhibitory effect of pemigatinib. Pemigatinib has no impact on the growth of parental Ba/F3 cells with $IC_{50}$ values exceeding 5 μM. It has great potency against the proliferation of FGFR1-3-translocated Ba/F3 cells ($IC_{50}$ of 1.2, 0.3 and 1.2 nM, respectively) but lower potency against the proliferation of FGFR4-translocated Ba/F3 cells ($IC_{50}$ of 12 nM), consistent with the results of kinase assays. Taken together, these results substantiate pemigatinib as a potent FGFR1-3 inhibitor.

**Structural basis of the pemigatinib-FGFR1 interaction**. To gain structural insights into the mechanism of FGFR inhibition by

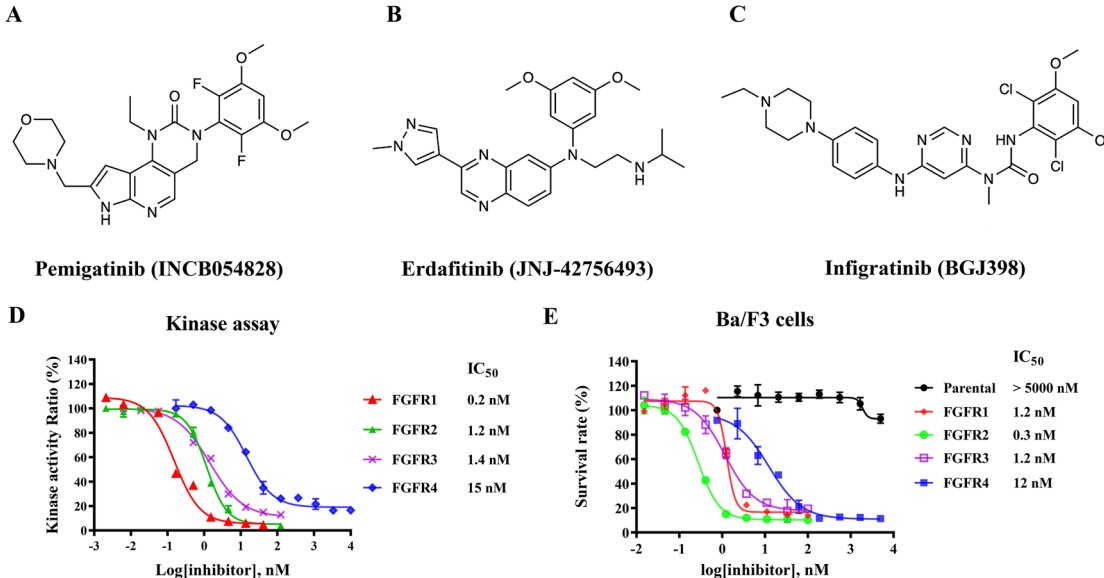

**Fig. 1 Pemigatinib is a potent FGFR1-3 inhibitor.** Chemical structures of pemigatinib (INCB054828) (**A**), Erdafitinib (JNJ-42756493) (**B**) and Infigratinib (BGJ398) (**C**). Inhibitory effects of pemigatinib against wild type FGFR1-4 using kinase activity inhibition assays (**D**) and Ba/F3 cell models expressing FGFR1-4 (**E**). Error bars represent the standard deviation for at least three independent measurements.

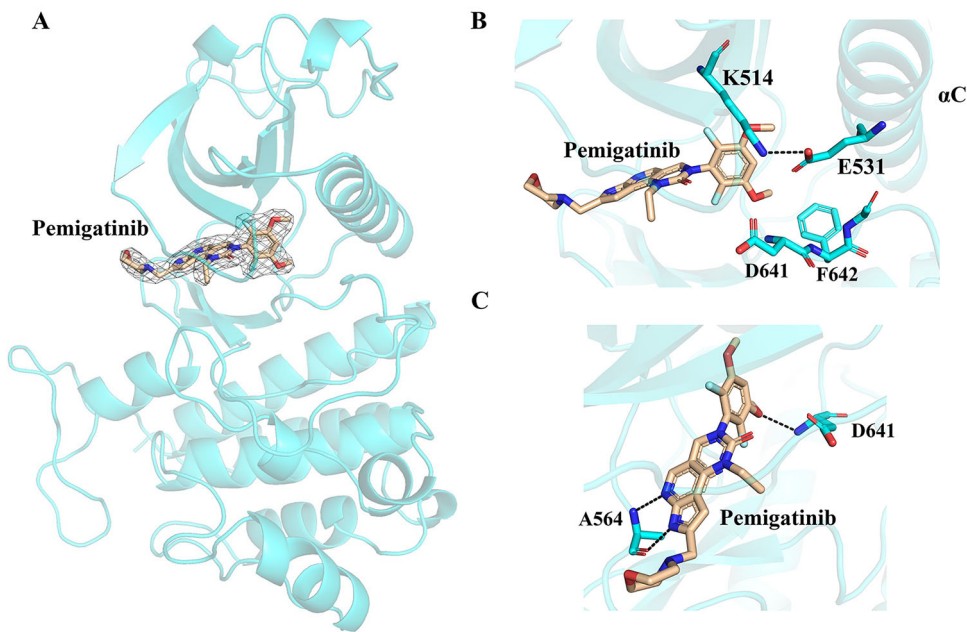

**Fig. 2 Crystal structure of pemigaitnib in complex with FGFR1. A** Overall structure of the pemigatinib/FGFR1 complex. **B** The activation loop of pemigaitnib kept in DFG-in conformation. **C** Hydrogen-bond interaction between pemigatinib and FGFR1.

pemigatinib, the crystal structure of the pemigatinib/FGFR1 kinase domain was determined at a high resolution of 2.5 Å. The detailed collection data are shown in Supplementary Table 1 and the validation report is available as Supplementary Data 1. The chemical structure of pemigatinib is described in Fig. 1A, and its electron density and structural characteristics are well represented in the crystal structure with the FGFR1 kinase domain (Fig. 2A). Pemigatinib occupies the ATP-binding pocket of FGFR1, where the activation loop adopts a DFG-in conformation (Fig. 2B), resembling other selective FGFR inhibitors. The pyrrolopyridine moiety forms two hydrogen bonds with the N-H group and the carbonyl group of Ala564 in the hinge region of FGFR1 (Fig. 2C). The difluoromethoxyphenyl ring is oriented perpendicular to the tricyclic scaffold and occupies the hydrophobic pocket containing the gatekeeper residue Val561. One of the methoxy oxygen atoms of the difluoromethoxyphenyl ring forms a hydrogen bond with the backbone nitrogen atom of Asp641 (Fig. 2C). The morpholine solubilizing group extends away from the hinge region toward the solvent exposed region and does not make any specific interactions with FGFR1. Apart from direct interactions, pemigatinib binds to FGFR1 indirectly via the water molecule that links the oxygen atom of the tricyclic scaffold to the amino side chain of Lys514 and the carboxylate side chain of Asp641 (Supplementary Fig. 1). More interactions, including van der Waals forces, are also described in Supplementary Fig. 1. These results demonstrate that pemigatinib firmly binds FGFR1 via an extensive interaction network.

**Structural comparison of pemigatinib to erdafitinib/infigratinib in complex with FGFR1.** To date, FGFR inhibitors approved by the FDA exclusively include pemigatinib, erdafitinib and infigratinib. To help fully understand the profiles of these three drugs, we performed a structural comparison of their binding to the FGFR1 kinase domain. The activation loops of these three drugs binding FGFR1 are all kept in the DFG-in conformation (Fig. 3A–C), indicating that they are classified as Type I inhibitors[24,25]. A 3,5-dimethoxyphenyl ring is observed in these three inhibitors, and one of the methoxy oxygen atoms is involved in a hydrogen bond with Asp641, which increases the

selectivity for FGFRs[8]. With regard to the molecular scaffold, the quinoxaline in erdafitinib formed one hydrogen bond with Ala564, whereas the tricyclic urea scaffold in pemigatinib and the N-aryl-N'-pyrimidin-4-yl urea scaffold in infigratinib formed two hydrogen bonds with Ala564 (Fig. 3D–F). More hydrogen bonds formed could stabilize the complex conformations, which may explain why pemigatinib (IC$_{50}$ of 0.2 nM) and infigratinib (IC$_{50}$ of 0.9 nM) have more potency toward FGFRs than erdafitinib (IC$_{50}$ of 1.2 nM)[25,26]. Moreover, they all possess a solubilizing group. As to pemigatinib, the solubilizing group is composed of a morpholine group, and to infigratinib and erdafitinib are ethyl piperazine and methyl pyrazole, respectively, which could increase their solubility and drug dissolution in the body[8,25]. Overall, these results indicate that the binding mode of pemigatinib in complex with FGFR1 is similar to that of erdafitinib/infigratinib.

**The effect of FGFR gatekeeper mutations on the potency of pemigatinib.** The question of whether gatekeeper mutations in the FGFR kinase domain influence on the inhibitory efficacy of pemigatinib has not been addressed. Thus, kinase activity inhibition assays were carried out to assess the sensitivity of pemigatinib to typical and frequent gatekeeper mutations, involving FGFR1 V561M, FGFR2 V564I/F and FGFR3 V555M (Fig. 4). Unexpectedly, pemigatinib retains excellent potency against FGFR2 V564I with an IC$_{50}$ of 7 nM in the kinase assay, which exhibits merely a 4.7-fold reduction in inhibitory efficacy compared to that against the wild type (Fig. 4A, B). In contrast, the efficacy of pemigatinib is evidently diminished by the introduction of methionine to replace valine. It is found that pemigatinib exhibits ~745-fold lower potency against FGFR1 V561M (IC$_{50}$ of 149 nM), and ~76-fold lower potency against FGFR3 V555M (IC$_{50}$ of 107 nM) (Fig. 4A, B). Moreover, pemigatinib is vulnerable to the V564F mutation in FGFR2. A great reduction in efficacy for pemigatinib is observed in the V564F mutation, with an IC$_{50}$ of 263 nM (~219-fold lower potency than FGFR1) (Fig. 4A, B). Likewise, decreased potencies against V564F and V555M are also observed in the cellular proliferation assay in Ba/F3 cells (Supplementary Fig. 2). These results reveal that

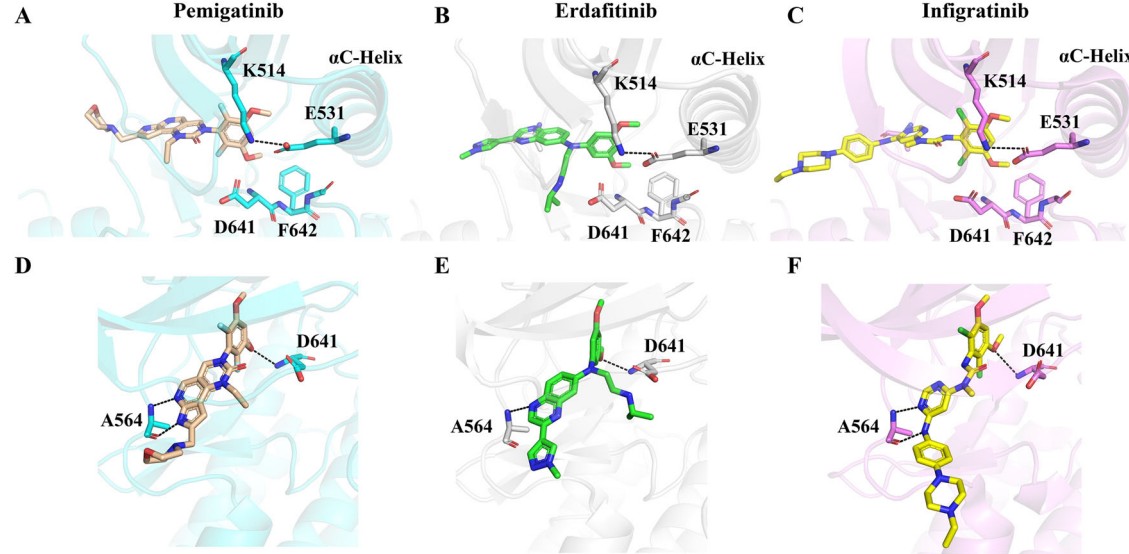

**Fig. 3 Structural comparison of pemigatinib to erdafitinib and infigratinib in complex with FGFR1. A–C** The DFG motif conformation of FGFR1. FGFR1 adopts a DFG-in activation loop conformation with pemigatinib, erdafitinib (PDB: 5EW8) and infigratinib (PDB: 3TT0). **D–F** Hydrogen-bond interactions between FGFR1 and pemigatinib/erdafitinib/infigratinib.

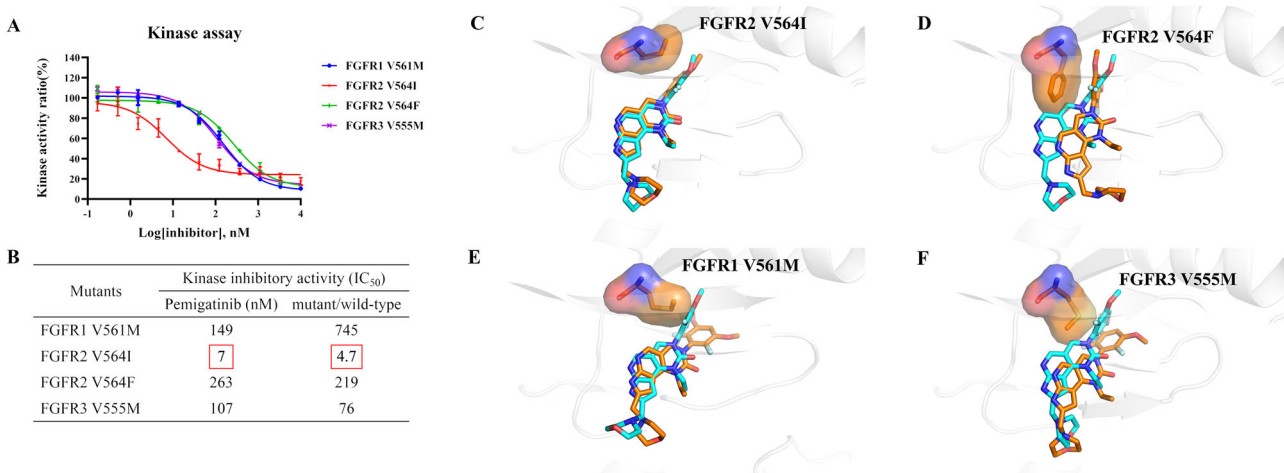

**Fig. 4 Pemigatinib remains the excellent inhibitory efficacy for Val-to-Ile gatekeeper mutation but lower potency against Val-to-Met/Phe gatekeeper mutation. A** Inhibitory effects of pemigatinib against FGFR1-3 gatekeeper mutants using kinase activity inhibition assays. **B** The collective half maximal inhibitory concentration (IC$_{50}$) values for FGFR1-3 gatekeeper mutations summarized in detail. Error bars represent the standard deviation for at least three independent measurements. **C–F** Structural models of FGFR2 V564I/F, FGFR1 V561M and FGFR3 V555M in complex with pemigatinib. Pemigatinib colored cyan is from the binding models with the corresponding wild type where pemigatinib is docked into FGFR2 (PDB ID: 6LVL), and FGFR3 (PDB ID: 7DHL), and is aligned with the structural models of gatekeeper mutants. Pemigatinib colored orange is the predicted structure with gatekeeper mutants. The structural models of FGFR2 V564I/F, FGFR1 V561M and FGFR3 V555M in complex with pemigatinib are generated by substitution of gatekeeper residues on the basis of FGFR1 (PDB: 7WCL), FGFR2 (PDB: 6LVL) and FGFR3 (PDB: 7DHL) structures.

pemigatinib could exhibit significant inhibition against the Val-to-Ile mutation but attenuated potency against the Val-to-Met/Phe mutation in FGFRs.

**Structural basis of FGFR gatekeeper mutations in response to pemigatinib**. To further understand the characteristics of pemigatinib, we evaluated the dynamic binding of pemigatinib with other FGFR types by flexible molecular docking using AutoDock Tools program[27]. Pemigatinib was redocked into FGFR1 to evaluate the reliability of this method. Then, pemigatinib was docked into FGFR2 (PDB ID: 6LVL)[28], FGFR3 (PDB ID:7DHL)[29] and FGFR4 (PDB ID: 7F3M)[30], respectively. Structural models

predict that FGFR1-4 in complex with pemigatinib exhibit similar binding modes and similar interaction patterns (Supplementary Fig. 3), with low predicted binding energies (Supplementary Table 2).

Furthermore, we also evaluated the dynamic binding of pemigatinib with FGFR gatekeeper mutations by flexible molecular docking. Structural superposition of FGFR2 V564I/pemigatinib and wild type FGFR2/pemigatinib indicates that pemigatinib binds with FGFR2 V564I in the similar position with wild type FGFR2 (Fig. 4C, Supplementary Fig. 4A and Supplementary Table 2). The less severe hindrance from the small isoleucine side chain makes the binding conformation nearly identical with the wild type, which underlies the sensitivity of pemigatinib toward FGFR2 V564I. When docked into

FGFR1 V561M, FGFR2 V564F and FGFR3 V555M, pemigatinib moderately moves out of the binding pocket in order to evade the steric clash with the bulky methionine and phenylalanine residues compared with their wild type (Fig. 4D–F). The movement of pemigatinib in these mutants may result in impaired inhibitory potencies and lower predicted binding energies ($-6.53$ Kcal·mol$^{-1}$ for FGFR1 V564M, $-8.65$ Kcal·mol$^{-1}$ for FGFR2 V564F and $-9.29$ Kcal·mol$^{-1}$ for FGFR3 V555M) (Fig. 4D–F, Supplementary Fig. 4B–D and Supplementary Table 2). Taken together, these structural models suggest that pemigatinib-FGFR binding is more susceptible to the disruption by the introduction of Met/Phe than Ile, due to the more severe steric clash caused by the bulky Met/Phe. Thus, the less severe hindrance could preserve excellent inhibitory potency against gatekeeper mutations.

## Discussion

Pemigatinib is the first small molecule targeted drug to treat cholangiocarcinoma[12]. Despite slight loss of efficacy for FGFR4, pemigatinib has great potency against FGFR1-3, indicating that pemigatinib represents a potential therapy for other FGFR-driven malignancies. Related clinical trials have been undertaken to expand the application of pemigatinib[12]. Structural analysis shows that pemigatinib can be divided into three important chemical components according to their interactions with FGFR1. First, the 3,5-dimethoxylphenyl ring is essential for the selectivity for FGFR, which may be enhanced by the introduction of fluorine/chlorine[8]. Second, the tricyclic urea scaffold features hydrogen-bond interactions with the hinge region, which could serve as a template for further design and screening of drug structures of FGFR and/or other kinase receptor inhibitors. Third, the solubilizing morpholine group regulates the lipophilicity and contributes to excellent rat and cynomolgus monkey PK profiles as well as abrogation of time-dependent inhibition issue[8]. This extensive network of pemigatinib-FGFR1 contact reinforces the strength of drug binding and accounts for the high potency of pemigatinib, providing an exemplification for the further design and optimization of FGFR inhibitors with high selectivity and high potency.

Furthermore, the structural comparison of pemigatinib to erdafitinib and infigratinib in complex with FGFR1 highlights the importance of these three chemical components in drug design and optimization. The 3,5-dimethoxyphenyl ring, molecular scaffold and solubilizing group are all observed in these three drugs with extensive and indispensable interactions with FGFR1, which are essential for drug discovery and design. By overlaying the structures of these three compounds binding to FGFR1 and the modeled gatekeeper mutations, we found that the dimethoxyphenyl rings of erdafitinib and infigratinib were slightly away from gatekeeper residue Val561/Ile561, similar to that of pemigatinib, suggesting that erdafitinib and infigratinib might have excellent potency toward FGFR1 V561I as with pemigatinib (Supplementary Fig. 5A, B). For FGFR1 V561M/F (Supplementary Fig. 5C, D), these mutations may have a great impact on the response to pemigatinib, erdafitinib and infigratinib resulting from severe steric hindrance. These modeling data provide the structural basis for the fact that FGFR1 V561M/F and FGFR4 V550M confer resistance to erdafitinib, and FGFR1 V561M/F, FGFR2 V564F and FGFR3 V555M confer resistance to infigratinib[31–35].

Pemigatinib is the first-generation FGFR inhibitor. According to previous experience with tyrosine kinase inhibitors[35–38], resistance would occur due to the gatekeeper mutations, such as FGFR1 V561M, FGFR2 V564I/F, FGFR3 V555M and so on[39]. In our study, we found that pemigatinib can maintain excellent activity against the Val-to-Ile gatekeeper mutation, but exhibit significant attenuated potency toward the Val-to-Met/Phe gatekeeper mutation in FGFR, which may be attributed to the severe steric clash between the bulky Met/Phe and the 3,5-dimethoxyphenyl ring. Therefore, overcoming resistance may be achieved by substituting a smaller group for the 3,5-dimethoxyphenyl ring with a larger space to accommodate the bulky gatekeeper residues. Moreover, increasing the flexibility of molecular inhibitors may also take effect. In the discovery of pemigatinib, the prototype is cyclized and generates a tricyclic urea scaffold to rigidify the structure and restore the FGFR activity[8], which reduces the flexibility of the compound and makes it vulnerable to Val-to-Met/Phe gatekeeper mutation. Hence, increased inhibitor flexibility could allow multiple inhibitor binding models to better accommodate gatekeeper mutation as well.

Except for the gatekeeper mutations, there are some other mutations conferring drug resistance to the FGFR inhibitors. For example, FGFR2 M535I, M537I, I547V, N549K/H/S/T, E565A/G, L617M/V, K641N/R and K659E/M/N have been reported[40]. Among of these mutations, six mutations have been observed in the treatment of pemigatinib, including N549K/H, E565A, K659M, L617V and K641R[41]. When mapped in our structure, these mutants are located outside of the binding pocket and are sorted into three categories (Supplementary Fig. 6). N549, E565 and K641 act as a "molecular brake" and keep the kinase in an autoinhibited state[42]. Thus, mutations of any residue of the triad might result in the disengagement of the "brake" and constitutive activation of the kinase[42]. Mutation of K659 would mimic the action of A-loop tyrosine phosphorylation, thereby constitutively activating the kinase[42]. L617M/V, M535I, M537I and I547V would strengthen the hydrophobic spine and stabilize the active kinase conformation, probably leading to less favorable pemigatinib-binding conditions[32,33,40]. In summary, these residues are all conserved in FGFR1-3, and any mutation of these residues might confer resistance to pemigatinib/erdafitinib/infigratinib by stabilizing the active kinase conformation either through disengaging the molecular brake or strengthening the hydrophobic spine or mimicking the action of A-loop tyrosine phosphorylation.

In this study, we performed kinase activity inhibition assays and cellular proliferation assays to demonstrate that pemigatinib is a potent and selective FGFR1–3 inhibitor. The structural analysis provides the features and nature of the pemigatinib-FGFR1 binding pattern, accounting for the high potency of pemigatinib. Pemigatinib could sustain the superior potency against the Val-to-Ile gatekeeper mutation than that against the Val-to-Met/Phe gatekeeper mutation, suggesting that the administration of pemigatinib at earlier stages of disease may evade Val-to-Ile mutation-based resistance. Collectively, these structural and inhibitory profiles exemplified by pemigatinib may be applied to the exploration of next-generation FGFR inhibitors to overcome gatekeeper mutation-based resistance.

## Methods

**Plasmid construction.** Human FGFRs were prepared as previously described[43–46]. Briefly, the kinase domains of FGFR1 (residues 458–765), FGFR2 (residues 453–770) and FGFR3 (residues 450–758) were respectively cloned into a modified pET28a expression vector with an N-terminal 6×His tag followed by a PreScission cleavage site. The mutant plasmids, FGFR1 C584S, FGFR1 V561M, FGFR2 V564I, FGFR2 V564F and FGFR3 V555M, were constructed by PCR using primers with the desired mutations.

**Protein expression and purification.** Plasmids were expressed in *E. coli* BL21 Rosetta cells. For FGFR1 C584S, Rosetta cells were co-expressed with YOPH to obtain non-phosphorylated proteins. The colony was inoculated into the liquid LB culture with 50 µg/mL kanamycin at 37 °C and induced at 16 °C for 18 h by the addition of 0.5 mM IPTG between OD$_{600nm}$ of 0.7~0.8. The cells were harvested by 15-min centrifugation at 3000 rpm. The pellets were resuspended in lysis buffer (20 mM Tris-HCl, pH 8.0, 500 mM NaCl, 20 mM imidazole, 0.5 mM TCEP) and

then lysed by a high-pressure homogenizer. The lysate supernatants were obtained after 35 min centrifugation at 18000 rpm and incubated with equilibrated Ni-NTA beads (GE Healthcare) for 1.5 h. Then, the beads were loaded into a gravity flow column and washed with lysis buffer containing 50 mM imidazole. The target proteins were eluted with lysis buffer containing 250 mM imidazole and digested with PreScission protease overnight to remove the N-terminal 6×His tag. Untagged FGFRs were further purified by anion exchange chromatography (GE Healthcare), and peak fractions collected were concentrated to 5–16 mg/mL. For crystallization, the pooled FGFR1 C584S was put through a Superdex 200 column (GE Healthcare) in storage buffer (20 mM Tris-HCl, pH 8.0, 20 mM NaCl, 0.5 mM TCEP).

**Kinase inhibition assay**. The kinase assays were performed using the ADP-Glo (Promega) methodology. Proteins (0.025–0.2 µM), pemigatinib (triple dilution method), ATP (10 µM) and poly (4:1 Glu, Tyr) peptides (Abcam, UK) (10 µM) were diluted with optimized kinase buffer (40 mM Tris-HCl pH 7.5, 20 mM MgCl$_2$, 20 mM NaCl, 0.1 mg/mL BSA, 1 mM TCEP, and 4% DMSO). The specific steps were prepared as previously described[47]. In short, the kinase reactions were initiated by mixing protein, inhibitor, ATP, and poly (4:1 Glu, Tyr) peptides, reacting at room temperature for 30 min, and then terminated by adding ADP-Glo after 40 min of incubation. Kinase activities were detected on a plate reader (Perkin Elmer) after the addition of detection reagent. IC$_{50}$ values were calculated with log[Inhibitor] versus kinase activity ratio using GraphPad Prism software.

**Crystallization and structure determination**. The hanging drop vapor diffusion method was used for crystallization as previously described[48,49]. FGFR1/pemigatinib crystals were produced by micro-seeding FGFR1 crystals. The FGFR1 crystals were grown at 4 °C by mixing 0.8 µL of protein solution with 0.8 µL of crystallization buffer comprising 18% (w/v) PEG 8000, 0.2 M LiSO4, and 0.1 M MES (pH 6.5). After 1 week, the crystals of FGFR1 were broken into microcrystals. FGFR1/pemigatinib crystals were produced similarly to the apo FGFR1 crystals but with the inhibitor molecule pre-incubated with FGFR1 in a 1:2 ratio overnight at 4 °C before micro-seeding and with identical crystallization conditions as apo FGFR1. The FGFR1/pemigatinib crystals were cryoprotected in the buffer supplemented with 20% glycerol and then flash-cooled in liquid nitrogen prior to data collection.

The X-ray diffraction data were collected in our lab, with HKL3000 employed for data integration and scaling[50]. The initial structures were solved by molecular replacement by Phaser from the PHENIX package with a search model from PDB entry 4RWJ[51]. Further structure refinement and model building were performed by Phenix. refine and Coot[50]. Full details are described in Supplementary Table 1. PyMOL and LigPlot+ were used for structural descriptions and protein–ligand interaction representations, respectively[52,53]. The coordinates and structure factors have been deposited in the PDB under accession numbers 7WCL.

**Ba/F3 cell line proliferation assays**. Cell proliferation and viability were evaluated in parental Ba/F3 cells and TEL-FGFR1-4/FGFR mutation-translocated Ba/F3 cell lines[54]. Cells were seeded in 96-well plates at $2 \times 10^3$ cells/well and incubated with various concentrations of pemigatinib in a final volume of 100 µL for 72 h at 37 °C. Subsequently, Cell Counting Kit-8 (Vazyme, China) was added, and after a 2-h incubation, the absorbance was measured at 450 nm using a multimode plate reader (Perkin Elmer). Each assay was performed in triplicate, and the IC$_{50}$ values were calculated for the inhibitory potency of pemigatinib in vivo by GraphPad Prism 7.0.

**Molecular docking**. Computational docking was performed to predict the binding of pemigatinib to FGFR2-4 and FGFR gatekeeper mutations. Pemigatinib was docked into FGFR2 (PDB ID: 6LVL), FGFR3 (PDB ID:7DHL) and FGFR4 (PDB ID: 7F3M), respectively. The structures of FGFR gatekeeper mutations were prepared by protein mutagenesis based on their corresponding wild type FGFR. The waters were eliminated and the polar hydrogen was added by PyMOL[53]. Gasteiger charges and rotatable bonds to pemigatinib were assigned by AutoDock Tools program[27]. The docking simulations were conducted with a flexible protein and a flexible ligand. A docking grid with the dimensions of 40*40*40 points in the x-, y-, and z-axis directions was built, which encompassed the entire ligand-binding clefts.

**Reporting summary**. Further information on research design is available in the Nature Research Reporting Summary linked to this article.

## Data availability

The coordinates and structure factors are deposited in the Protein Data Bank under the accession codes 7WCL (FGFR1/pemigatinib complex). Validation report is available as Supplementary Data 1. All other relevant data supporting the key findings of this study are available within the article and its Supplementary Information files. A reporting summary is available as a Supplementary Information file.

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

## Acknowledgements

The present study was financially supported by the Natural Science Foundation of China (No. 81570537, 81974074 and 82172654), Hunan Provincial Science and Technology Department (2018RS3026 and 2021RC4012) and National Science Foundation of Hunan Province (grants 2021JJ40961 and 2022JJ40763). We thank the staff from the Shanghai Synchrotron Radiation Facility (SSRF) beamline BL17U/BL19U/BL02U/BL10U/BL17B for assistance during diffraction data collection.

## Author contributions

Q.L. and X.C. performed experiments; L.Q. and M.G. performed data collection and structure determination. H.W. and S.D. analyzed the data. Q.L., X.C., L.J. and Y.C. prepared the manuscript.

## Competing interests

The authors declare no competing interests.
