## [Peer Review File · Communications Chemistry]

Reviewers' comments:

Reviewer #1 (Remarks to the Author):

The manuscript by Lin et al. provides the structural basis for the binding of pemigatinib to fibroblast growth factor receptor 1 (FGFR1) and characterized the effects of mutating the gatekeeper residue (Val561) on pemigatinib binding. This is a solid study that shed insight into the binding mechanism of a clinical drug for the first time, and generate useful predictions on clinical resistance to this targeted therapy. Below are some detailed comments.

1. Figure legends seems to be missing!
2. The biochemical IC50 values indicate pemigatinib binds to FGFR1 most strongly (with 6-, 7-, 75-fold selectivity over FGFR2/FGFR3/FGFR4 respectively). Can the authors explain the differences based on their FGFR1 co-crystal structure?
3. The authors should discuss insight gained from their study and possible strategies that could overcome V561M or V561F mutations
4. spell out FGFR at its first appearance in Abstract.

Reviewer #2 (Remarks to the Author):

In this work, the authors undertake biochemical and structural analysis on pemigatinib against FGFRs as well as gatekeeper mutations. Results showed that except for FGFR1-3, Pemigatinib is also potent against Val-to-Ile gatekeeper mutation but cannot overcome Val-to-Met/Phe gatekeeper mutation in FGFR. The discovery may be of some interest to Communications Chemistry. However, there are some issues need to be further addressed.

- 1) As the authors only determined the crystal structure of pemigatinib/FGFR1 kinase domain, it is essential to evaluate the dynamic binding of pemigatinib with other FGFR type as well as their gatekeeper mutations using some computational methods such as molecular dynamics or flexible molecular docking.
- 2) It is also interesting to test whether pemigatinib shows great potency against the proliferation of FGFR1-3 gatekeeper mutations-translocated Ba/F3 cells.
- 3) The authors conducted structural comparison of pemigatinib to erdafitinib/infigratinib in complex with WT FGFR1. However, the focus of this work is about the gatekeeper mutation resistance of pemigatinib. And it has been reported that erdafitinib/infigratinib also show resistance to FGFR gatekeeper mutations (see Yue, S.; Li, Y.; Chen, X.; Wang, J.; Li, M.; Chen, Y.; Wu, D. FGFR-TKI resistance in cancer: current status and perspectives. *J Hematol Oncol* 2021, 14, 23). So it is import to compare their binding mode with gatekeeper-mutated FGFR and summarize the possible reasons.
- 4) Except for the gatekeeper mutations, there are some other mutations conferring drug resistance to the FGFR inhibitors (also see the reference paper for question 3), please also comment on this.
- 5) As pemigatinib is already known as a selective and potent FGFR inhibitor, please try to revise your manuscript title to better represent the novelty of your work.

Response to Reviewers' Comments

Reviewer #1 (Remarks to the Author):

The manuscript by Lin et al. provides the structural basis for the binding of pemigatinib to fibroblast growth factor receptor 1 (FGFR1) and characterized the effects of mutating the gatekeeper residue (Val561) on pemigatinib binding. This is a solid study that sheds insight into the binding mechanism of a clinical drug for the first time, and generates useful predictions on clinical resistance to this targeted therapy. Below are some detailed comments.

1. Figure legends seems to be missing!

Response: Thank you for your review and reminding. We apologize for this mistake, and the figure legends have been added to the revised manuscript.

2. The biochemical IC₅₀ values indicate pemigatinib binds to FGFR1 most strongly (with 6-, 7-, 75-fold selectivity over FGFR2/FGFR3/FGFR4 respectively). Can the authors explain the differences based on their FGFR1 co-crystal structure?

Response: Thank you for your review and comment. We may not explain the differences based only on our crystal structure. To solve this problem, we built several structural models of other FGFR in complex with pemigatinib by flexible molecular docking. As shown in new Figure S3 and Table S2, the binding modes of FGFR1-4 in complex with pemigatinib share high structural similarity and low predicted binding energies. The inhibitory differences may depend on other factors, such as some subtle conformational changes outside the ATP pocket of kinase domain resulted from the protein-drug binding.

3. The authors should discuss insight gained from their study and possible strategies that could overcome V561M or V561F mutations

Response: Thank you for your comments and advice.

In this paper, we found in pemigatinib that: 1) an extensive interaction network in the pemigatinib-FGFR1 binding mode that accounts for the high potency of pemigatinib; 2) three chemical components (3,5-dimethoxyphenyl ring, molecular scaffold and solubilizing group) that play important and dispensable roles in the drug-protein interactions; and 3) the response to the Val-to-Ile/Met/Phe gatekeeper mutation that facilitates the development of the next-generation FGFR inhibitors to overcome gatekeeper mutations.

To provide a deeper insight into our research, we propose some possible strategies to overcome gatekeeper mutations based on our structural and functional analysis upon pemigatinib. We have added these possible strategies to the Discussion section in the revised manuscript.

4. spell out FGFR at its first appearance in Abstract.

Response: Thank you for your reminding. We have modified and spelled out “FGFR” at its first appearance in the Abstract.

Reviewer #2 (Remarks to the Author):

In this work, the authors undertake biochemical and structural analysis on pemigatinib against FGFRs as well as gatekeeper mutations. Results showed that except for FGFR1-3, Pemigatinib is also potent against Val-to-Ile gatekeeper mutation but cannot overcome Val-to-Met/Phe gatekeeper mutation in FGFR. The discovery may be of some interest to Communications Chemistry. However, there are some issues need to be further addressed.

1. As the authors only determined the crystal structure of pemigatinib/FGFR1 kinase domain, it is essential to evaluate the dynamic binding of pemigatinib with other FGFR type as well as

their gatekeeper mutations using some computational methods such as molecular dynamics or flexible molecular docking.

Response: Thank you for your comments and advice. We have evaluated the dynamic binding of pemigatinib with other FGFR types as well as their gatekeeper mutations by flexible molecular docking and added redrawn Figure 4, new Figure S4 and Table S2 to the revised manuscript.

2. It is also interesting to test whether pemigatinib shows great potency against the proliferation of FGFR1-3 gatekeeper mutations-translocated Ba/F3 cells.

Response: Thank you for your comments and advice. We have performed cell proliferation assays to investigate the inhibition of pemigatinib in response to gatekeeper mutations, and shown in new Figure S2.

3. The authors conducted structural comparison of pemigatinib to erdafitinib/infigratinib in complex with WT FGFR1. However, the focus of this work is about the gatekeeper mutation resistance of pemigatinib. And it has been reported that erdafitinib/infigratinib also show resistance to FGFR gatekeeper mutations (see Yue, S.; Li, Y.; Chen, X.; Wang, J.; Li, M.; Chen, Y.; Wu, D. FGFR-TKI resistance in cancer: current status and perspectives. J Hematol Oncol 2021, 14, 23). So it is import to compare their binding mode with gatekeeper-mutated FGFR and summarize the possible reasons.

Response: Thank you for your comments and advice. We have conducted a structural comparison of pemigatinib to erdafitinib/infigratinib in complex with FGFR1-3 gatekeeper mutations (Figure S5) and summarized the possible reasons in the discussion section in the revised manuscript. Val-to-Met/Phe gatekeeper mutation may have a greater impact on the potencies of pemigatinib, erdafitinib and infigratinib than Val-to-Ile gatekeeper mutation, attributing to the severe steric hindrance from Met/Phe residue.

4. Except for the gatekeeper mutations, there are some other mutations conferring drug resistance to the FGFR inhibitors (also see the reference paper for question 3), please also comment on this.

Response: Thank you for your comments and advice. We have mapped these clinically related mutations in our structure and commented on these drug resistance mutations in the revised manuscript.

5. As pemigatinib is already known as a selective and potent FGFR inhibitor, please try to revise your manuscript title to better represent the novelty of your work.

Response: Thank you for your comments and advice. We have revised our manuscript title to “Characterization of pemigatinib against FGFR gatekeeper mutants”.

REVIEWERS' COMMENTS:

Reviewer #1 (Remarks to the Author):

The revision has addressed my prior comments satisfactorily.

Reviewer #2 (Remarks to the Author):

The authors have addressed all the comments. Now it can be published.